# Consistent Treatment of Quantum Systems with a Time-Dependent Hilbert Space

**DOI:** 10.3390/e26040314

**Published:** 2024-04-03

**Authors:** Ali Mostafazadeh

**Affiliations:** Departments of Mathematics and Physics, Koç University, Sarıyer, 34450 Istanbul, Turkey; amostafazadeh@ku.edu.tr

**Keywords:** time-dependent Hilbert space, time-dependent inner product, gauge theory, Hermitian vector bundle, Hilbert bundle, pseudo-Hermitian operator

## Abstract

We consider some basic problems associated with quantum mechanics of systems having a time-dependent Hilbert space. We provide a consistent treatment of these systems and address the possibility of describing them in terms of a time-independent Hilbert space. We show that in general the Hamiltonian operator does not represent an observable of the system even if it is a self-adjoint operator. This is related to a hidden geometric aspect of quantum mechanics arising from the presence of an operator-valued gauge potential. We also offer a careful treatment of quantum systems whose Hilbert space is obtained by endowing a time-independent vector space with a time-dependent inner product.

## 1. Introduction

Quantum mechanics is perhaps the most successful physical theory of all times. Its basic principles and mathematical foundation have been known since the 1930s and covered in numerous textbooks and monographs written over the past nine decades [1,2,3,4,5,6]. Yet the discovery of non-Hermitian PT symmetric Hamiltonians possessing a real spectrum towards the end of 20th century [7,8,9] created the impression among some physicists that there were gaps in our understanding of the fundamental aspects of quantum mechanics [10,11,12]. This motivated the reexamination of some of the basic issues related to the necessity of some of the postulates of quantum mechanics [13,14,15,16] and led to the confirmation of the fact that such Hamiltonians could define a unitary quantum dynamics provided that we could uphold each and every one of the postulates of quantum mechanics [2] by an appropriate redefinition of the Hilbert space of the system [17,18,19,20,21].

Among the benefits of studying non-Hermitian PT symmetric Hamiltonians was the development of methods for defining invariant inner products [13,14,21,22] and using these to address some old and important problems related to the construction of the Hilbert space and basic observables in quantum cosmology [23,24] and relativistic quantum mechanics of scaler [25,26,27] and spin-1 [28,29,30] particles. Quantum cosmological applications of these methods yield quantum systems with a Hilbert space whose inner product depends on the logarithm of the scale factor, which is a time-like degree of freedom [23,24]. This has motivated the study of the role of the Hamiltonian operator in a Hilbert space with a time-dependent inner product and led to the discovery of a quantum mechanical analog of the principle of general covariance of general relativity [31]. It has also revealed a conflict between the observability of the Hamiltonian and the unitarity of the dynamics it generates [32]. Different authors have proposed ways to deal with this conflict [33,34,35,36,37,38,39]. Among these is a geometric resolution that allows for a genuine geometric extension of quantum mechanics [40] and uncovers a previously unnoticed geometric aspect of quantum mechanics [41].

In the present article, we provide a careful and essentially self-contained discussion of some of the basic problems related to the formulation of the kinematics and dynamics of quantum systems and offer a thorough treatment of quantum systems defined on time-dependent Hilbert spaces. In particular, we consider systems whose Hilbert space acquires its time-dependence through the time-dependence of its inner product. This requires a detailed analysis of a dynamical generalization of the notion of pseudo-Hermiticity [13], which was initially employed in the context of quantum cosmology [23] and the quantum mechanical analog of the principle of general covariance [31]. It also plays a central role in the context of the conflict between observability of the Hamiltonian and the unitarity of the dynamics it generates [32,33,34,35,36,37,38,39,40,41].

We close this section by posing a simple question related to the quantum mechanical analogs of time-dependent canonical transformations of classical mechanics [42]. Consider a quantum system whose state vectors belong to a time-independent Hilbert space ℋ and whose dynamics are determined by the Schrödinger equation,
(1)$$i\hbar \frac{d}{dt}\psi \left(t\right)=H\psi \left(t\right),$$
where *H* is a time-independent Hermitian Hamiltonian operator. The observables of the system are represented by Hermitian operators *O* acting in ℋ. Quantum mechanical analogs of the classical canonical transformations are unitary transformations of the state vectors,
(2)$$\phi ⟶\tilde{\phi }:=U\left(t\right)\phi ,$$
under which the observables *O* and the Hamiltonian H(t) transform according to
(3)$$\begin{array}{cc} \; & O⟶\tilde{O}:=U\left(t\right)O\;U{\left(t\right)}^{-1}, \end{array}$$
(4)$$\begin{array}{cc} \; & H\left(t\right)⟶\tilde{H}\left(t\right):=U\left(t\right)H\;U{\left(t\right)}^{-1}+i\hbar \left[\frac{d}{dt}U\left(t\right)\right]U{\left(t\right)}^{-1}. \end{array}$$

By virtue of the fact that U(t):ℋ→ℋ is a unitary operator, (Equation 2) and (Equation 3) ensure the invariance of the expectation values of the observables, i.e.,
$$\frac{\langle \phi |O\phi \rangle }{\langle \phi |\phi \rangle }⟶\frac{\langle \tilde{\phi }|\tilde{O}\tilde{\phi }\rangle }{\langle \tilde{\phi }|\tilde{\phi }\rangle }=\frac{\langle \phi |O\phi \rangle }{\langle \phi |\phi \rangle },$$
while (Equation 4) guarantees that the transformation (Equation 2) maps the solutions of the Schrödinger (Equation 1) to those of
(5)$$i\hbar \frac{d}{dt}\tilde{\psi }\left(t\right)=\tilde{H}\left(t\right)\tilde{\psi }\left(t\right),$$
where $\tilde{\psi }\left(t\right):=U\left(t\right)\psi \left(t\right)$.

The unitarity of the transformation (Equation 2) and the transformation rules (Equation 3) and (Equation 4) imply that the physical quantities that quantum mechanics allows us to compute are not affected by this transformation. This is standard textbook material, but it leads to the following dilemma. Usually we take the Hamiltonian to represent the energy operator and identify the energy levels of the system with points in the spectrum of the Hamiltonian. But the transformation rule for the Hamiltonian, namely (Equation 4), does not leave its spectrum invariant. Therefore, even if we take the Hamiltonian *H* to be the energy observable, we cannot claim that the transformed Hamiltonian $\tilde{H}\left(t\right)$ represents the energy observable, because its spectrum can be completely different from that of *H*. For example if we identify U(t) with the inverse of the time-evolution operator, i.e., set U(t)=exp(itH/ℏ), which we do in going from the Schrödinger picture of dynamics to its Heisenberg picture [4], we find that $\tilde{H}\left(t\right)$ vanishes identically. Hence its spectrum is {0}. A simple way of avoiding this problem is to identify the energy operator before the unitary transformation and use the transformation rule for the observables, namely (Equation 3), to determine the energy observable after the transformation.

As we explain below, a quantum system is uniquely determined by the choice of its Hilbert space and Hamiltonian operator. But this choice is not unique, for all Hilbert space–Hamiltonian pairs that are related by unitary transformations provide physically equivalent descriptions of the system. Therefore, one must not be able to distinguish between different choices of such pairs. Yet the above prescription of identifying the energy observable seems to assume the existence of a special Hilbert space–Hamiltonian pair, which allows us to identify the energy observable with the Hamiltonian. This raises the natural question: “How do we find this special pair?” One of the aims of the present article is to offer a general prescription for determining the energy observable that is independent of the choice of the description of the system in terms of the Hilbert space–Hamiltonian pairs.

## 2. Basic Facts about Quantum Kinematics and Dynamics

Every quantum system can be described by a Hilbert space ℋ and a linear operator *H* acting in ℋ called the Hamiltonian. Throughout this article we use the term “Hilbert space” to mean a complex separable Hilbert space, i.e., a complex vector space endowed with a positive-definite inner product such that as an inner-product space it is complete (i.e., the Cauchy sequences converge) and has countable orthonormal bases [43].

The (pure) states of the system are identified with the one-dimensional subspaces (rays) of ℋ, and its observables are given by certain linear operators acting in ℋ. Each nonzero element ψ of ℋ determines a unique state of the system, namely {αψ|α∈C}. For this reason we call nonzero elements of ℋ “state vectors” and label states of the system by ${𝒮}_{\psi }$, where ψ is a state vector belonging to ${𝒮}_{\psi }$. Clearly ${𝒮}_{\phi }={𝒮}_{\psi }$, if and only if ϕ=αψ for some nonzero complex number α.

The central ingredient of quantum mechanics that distinguishes it from classical theories is its measurement or projection axiom. According to this axiom, quantum mechanics does not allow for the prediction of the outcome of a measurement even if we have complete information about the observable we measure and the state of the system before the measurement. We can only use quantum mechanics to determine possible outcomes of the measurement, the probabilities of measuring these outcomes, and their expectation (ensemble average) values. Both of the latter quantities have the form
(6)$$\frac{\langle \psi ,L\psi \rangle }{\langle \psi ,\psi \rangle },$$
where 〈·,·〉 denotes the inner product of ℋ, ψ is a state vector belonging to the domain of definition of *L* such that ${𝒮}_{\psi }$ is the state of the system immediately before the measurement, and *L* is either a projection operator associated with a subset S of the spectrum of the observable or the observable itself. In the former case, (Equation 6) gives the probability of measuring a value for the observable that belongs to S. In the latter case, it gives the expectation value of the observable. In both cases, and regardless of the choice of ψ, (Equation 6) is necessarily a real number. This simple requirement turns out to put a severe restriction on the operator *L*. Specifically, it implies that *L* must satisfy
(7)〈ϕ,Lψ〉=〈Lϕ,ψ〉for allϕ,ψ∈Dom(L),
where Dom(L) stands for the domain of *L*. We can use the basic properties of the inner product to show that (Equation 7) implies the realness of the right-hand side of (Equation 6) for all ψ∈Dom(L). Showing that the latter requirement implies (Equation 7) requires slightly more work ([Theorem 1.6.1] [44]).

Linear operators *L* fulfilling (Equation 7) are said to be “symmetric”. Some authors call them “Hermitian” [44]. But this is not consistent with the terminology adopted by von Neumann in his monumental book on quantum mechanics [2]. von Neumann uses the term “Hermitian” for what is nowadays called “self-adjoint” [43,44,45]. In the present article, we follow von Neumann’s terminology and use the terms “Hermitian” and “self-adjoint” interchangably. The precise definition of this concept which applies to finite- as well as infinite-dimensional Hilbert spaces is rather technical. We therefore present it in Appendix A where we also discuss the basic requirements that disqualify non-Hermitian symmetric operators to serve as observables for quantum systems with infinite-dimensional Hilbert spaces.

It is easy to see that (Equation 7) implies the realness of the eigenvalues of *L*. The converse of this statement is, however, not true, i.e., there are non-symmetric linear operators whose eigenvalues are real. For example, suppose that ℋ is the vector space $C^{2\times 1}$ of 2×1 complex matrices (column vectors) endowed with the standard Euclidean inner product,
$$\langle w,z\rangle :=w^{†}z,$$
where $w,z\in C^{2\times 1}$, and the superscript † stands for the Hermitian conjugate (complex conjugate of the transpose) of a matrix. It is an elementary fact of linear algebra that every linear operator $L:C^{2\times 1}\to C^{2\times 1}$ with domain $C^{2\times 1}$ is given by a 2×2 matrix L according to Lz=Lz. This, in particular, implies that the eigenvalues of *L* coincide with those of L, and $\langle w,Lz\rangle =w^{†}Lz$. Let us examine the following choices for L, w, and z.
$$\begin{array}{cccccc} \; & L:=\left[\begin{array}{cc} 1 & 0\\ -i & 2 \end{array}\right],\; & \; & w:=\left[\begin{array}{c} 1\\ 0 \end{array}\right],\; & \; & z:=\left[\begin{array}{c} 0\\ 1 \end{array}\right]. \end{array}$$

It is easy to see that the spectrum of L (and consequently of *L*) consists of real eigenvalues, 1 and 2. We also have
$$\begin{array}{cccc} \; & \langle w,Lz\rangle =w^{†}Lz=0,\; & \; & \langle Lw,z\rangle ={\langle z,Lw\rangle }^{*}={\left(z^{†}Lw\right)}^{*}=i. \end{array}$$Therefore, although *L* has a real spectrum, it violates (Equation 7). Notice also that the expectation value of *L* in the state ${𝒮}_{u}$ with u:=w+z fails to be real; a quick calculation gives
$$\frac{\langle u,Lu\rangle }{\langle u,u\rangle }=\frac{u^{†}Lu}{u^{†}u}=\frac{3-i}{2}.$$This is a clear proof that the realness of the eigenvalues of a linear operator does not ensure the realness of its expectation values.

Standard textbooks on quantum mechanics also use von Neumann’s terminology and identify observables of quantum systems with “Hermitian operators”. Most of them, however, do not discuss the difference between symmetric and Hermitian operators, and refer to condition (Equation 7) as the “Hermiticity condition”. This causes no difficulties when the Hilbert space ℋ of the system is finite-dimensional, because in this case the observables are defined everywhere in ℋ. As we explain in Appendix A, this makes the conditions of being symmetric and Hermitian equivalent. Therefore every non-Hermitian operator defined in a finite-dimensional Hilbert space violates (Equation 7) regardless of whether its eigenvalues are real or not.

The requirement of the realness of the eigenvalues of observables is a logical consequence of the fact that they are the possible readings of measuring devices one employs to measure these observables. This seems to have provided the basic motivation for using non-Hermitian operators with a real spectrum as Hamiltonian operators for certain quantum systems [7,10,12]. Because these operators violate (Equation 7), there are states in which their expectation values fail to be real. Therefore, they cannot be identified with the statistical averages of readings of a measuring device, which are necessarily real numbers. This argument also applies when ℋ is infinite-dimensional. The realness of the spectrum of a linear operator does not ensure the realness of its expectation values. This is a firmly established mathematical fact that is unfortunately not covered in a great majority of textbooks on quantum mechanics.

Because the knowledge of the Hilbert space ℋ is sufficient for the identification of its one-dimensional subspaces and Hermitian operators acting in ℋ, the states and observables of the quantum system, and consequently, its kinematic structure are determined by ℋ. In general, changing ℋ would drastically change the set of states and observables of the system. In particular, it might be possible to change the inner product on ℋ in such a way that the domain of a non-Hermitian linear operator remains unchanged but it acts as a Hermitian operator in the new Hilbert space. This is the basic idea of the pseudo-Hermitian representation of quantum mechanics [21], which was originally developed in an attempt to achieve the following seemingly unrelated objectives:To establish a precise mathematical framework [13,14,15] to study non-Hermitian PT-symmetric Hamiltonians [7] and elucidate their physical content [17,18,46];To address the problem of the construction of the Hilbert space and basic observables in minisuperspace quantum cosmology [23,24].

The information about the dynamical properties of a quantum system is contained in the Hamiltonian. In the Schrödinger picture of dynamics, the evolution of the state vectors ψ(t) of the system is governed by the requirement that they satisfy the Schrödinger equation,
(8)$$i\hbar \frac{d}{dt}\psi \left(t\right)=H\left(t\right)\psi \left(t\right).$$Here, we use the symbol H(t) for the Hamiltonian to reflect the fact that it may depend on time. In general, the Hamiltonian involves a set of real classical control parameters or coupling constants, $R_{1},R_{2},\cdots ,R_{d}$, whose values may change with time. These determine the time-dependence of the Hamiltonian according to H(t):=H[R(t)], where *R* stands for $\left(R_{1},R_{2},\cdots ,R_{d}\right)$. In general, we can identify the latter with (the coordinates of) points of a parameter space *M*, and view the function R(t) as a parameterized curve in *M*. A well-known example is a spin-*s* particle interacting with a rotating magnetic field B(t), where ℋ is $C^{2s+1}$ endowed with the Euclidean inner product, $\langle \psi ,\phi \rangle :=\sum_{j=1}^{2s+1}\psi_{j}^{*}\phi_{j}$,
$$H\left(t\right)=\omega_{L}\hat{B}\left(t\right)\cdot S,$$$\omega_{L}$ is the Larmor frequency, $\hat{B}\left(t\right):={\left|B\right|}^{-1}B\left(t\right)$ is the direction of B(t), and $S=(S_{x},S_{y},S_{z})$ is the spin operator in its standard 2s+1-dimensional unitary representation [47]. Because $\hat{B}\left(t\right)$ traces a curve on the unit sphere,
$$S^{2}:=\left\{\left(x,y,z\right)\in R^{3}\;|\;x^{2}+y^{2}+z^{2}=1\right\},$$
the parameter space of this system is $S^{2}$. If we identify the axis of rotation of the magnetic field with the *z* axis, we can express $\hat{B}\left(t\right)$ in terms of the azimuthal and polar spherical coordinates, φ and ϑ. This gives
$$\hat{B}\left(t\right)=\left(\sin\vartheta \cos\varphi (t),\sin\vartheta \sin\varphi (t),\cos\vartheta \right),\;\;\;\varphi \left(t\right)=\omega t,$$
where ω is the angular speed of rotation of the magnetic field about the *z* axis. These formulas suggest taking $R_{1}:=\varphi$ and $R_{2}:=\vartheta$ as the parameters of the system.

In the standard textbook description of quantum mechanics, the kinematical structure of the system is independent of its dynamics. This is because the Hilbert space ℋ does not depend on time. There are, however, situations where this is not the case. A simple example is a non-relativistic particle trapped in an infinite square well potential with a time-dependent width w(t) in one dimension [48,49,50,51], i.e.,
(9)$$v\left(x,t\right):=\left\{\begin{array}{ccc} 0 & \mathrm{for} & x\in [0,w(t)],\\ \infty & \mathrm{for} & x\notin [0,w(t)]. \end{array}\right.$$

The Hilbert space of this system is the space of square-integrable functions defined on the interval [0,w(t)], i.e., $ℋ=ℋ\left(t\right)=L^{2}[\left(0,w\left(t\right)\right]$. Consider an evolving state vector ψ(t) for this system. At different instants of time, ψ(t) belongs to different Hilbert spaces. This makes the very definition of its time derivative, namely
(10)$$\frac{d}{dt}\;\psi \left(t\right):=\lim_{\epsilon \to 0}\frac{\psi (t+\epsilon )-\psi (t)}{\epsilon },$$
problematic; because ψ(t+ϵ)∈ℋ(t+ϵ) while ψ(t)∈ℋ(t), it is not meaningful to speak of ψ(t+ϵ)−ψ(t). This is a serious problem as the Schrödinger equation (Equation 8) involves $\frac{d}{dt}\;\psi \left(t\right)$.

Fortunately, it is possible to devise an alternative description of this system that makes use of a time-independent Hilbert space [51]. This description avoids the problem with the definition of $\frac{d}{dt}\;\psi \left(t\right)$ when ψ(t) belongs to a time-dependent Hilbert space ℋ(t), but it does not offer a general solution for it.

There is a class of quantum systems described by a time-dependent Hilbert space ℋ(t) where the set of state vectors and the rules according to which they are added and multiplied by numbers do not change with time. This means that the vector-space structure of ℋ(t) is time-independent. The time-dependence of such a Hilbert space stems from the time-dependence of its inner product. For this class of quantum systems, the term $\frac{1}{\epsilon }\left[\psi \left(t+\epsilon \right)-\psi \left(t\right)\right]$ is meaningful, but (Equation 10) is still unacceptable, because the operation of evaluating the ϵ→0 limit in its right-hand side is ill-defined. Recall that given a function *f* mapping R to a Hilbert space ℋ, $\xi :=\lim_{\epsilon \to 0}f\left(\epsilon \right)$ means that there is some ξ∈ℋ such that $\lim_{\epsilon \to 0}\left\|f\left(\epsilon \right)-\xi \right\|=0$, where ‖·‖ stands for the norm of the Hilbert space. To make sense of the right-hand side of (Equation 10), we need a unique choice for the norm of $\frac{1}{\epsilon }\left[\psi \left(t+\epsilon \right)-\psi \left(t\right)\right]-\frac{d}{dt}\;\psi \left(t\right)$, which is unavailable because ψ(t+ϵ) and ψ(t) belong to Hilbert spaces with different inner products.

## 3. Quantum Dynamics in a Time-Dependent Hilbert Space

Consider a quantum system described by a time-dependent Hilbert space ℋ(t) and a possibly time-dependent Hamiltonian operator H(t). In analogy with the description of time-dependent Hamiltonians that we offer in the preceding section, we can imagine that ℋ(t) acquires its time-dependence through its dependence on a set of time-dependent real parameters, $R_{1},R_{2},\cdots ,R_{d}$, i.e., ℋ(t)=ℋ[R(t)] with $R:=(R_{1},R_{2},\cdots ,R_{d})$. Again, we can identify these parameters with coordinates of points of a parameter space *M* in some coordinate system, so that as time progresses R(t) traces a curve in *M*. This description of a quantum system with a time-dependent Hilbert space involves the assignment of a Hilbert space ℋ[R] to each point *R* of *M*. Imposing the natural requirement that the dimension *N* of ℋ[R] (which can be finite or infinite) be independent of *R*, we arrive at a bundle of Hilbert spaces attached to points of *M*. This is an example of what mathematicians call a Hermitian vector bundle (Hilbert bundle when N=∞) [52,53]. This suggests that we should replace the role of the time derivative of the evolving state (Equation 10), which applies to dynamics taking place in a constant Hilbert space, with an appropriate notion of covariant time derivative on a Hermitian vector bundle E, [52,54,55].

### 3.1. Covariant Differentiation with Respect to Time

Consider an orthonormal basis ${\left\{\phi_{n}\left[R\right]\right\}}_{n=1}^{N}$ of ℋ[R] for each *R*, and suppose that $\phi_{n}\left[R\right]$ are smooth functions of *R* at least in some open subset ${𝒪}_{\alpha }$ of *M* where our coordinate system is defined. To be precise, *M* is a smooth manifold and ${𝒪}_{\alpha }$’s provide an open covering of *M* by coordinate charts [54,55]. Given an evolving state vector ψ(t) that belongs to ℋ[R(t)], we can expand it in ${\left\{\phi_{n}\left[R\left(t\right)\right]\right\}}_{n=1}^{N}$. This gives
(11)$$\psi \left(t\right)=\sum_{n=1}^{N}c_{n}\left(t\right)\;\phi_{n}\left[R\left(t\right)\right],$$
where
(12)$$c_{n}\left(t\right):={\langle \phi_{n}\left[R\left(t\right)\right],\psi \left(t\right)\rangle }_{R(t)},$$
and ${\langle \cdot ,\cdot \rangle }_{R}$ is the inner product of ℋ[R]. The basis expansion (Equation 11) applies for the values of *t* such that $R\left(t\right)\in {𝒪}_{\alpha }$.

Now, suppose that ℋ is a fixed (separable) Hilbert space having the same dimension as ℋ[R], namely *N*. Because separable Hilbert spaces with the same dimension are isomorphic, there are unitary operators mapping ℋ to ℋ[R]. To specify one, we choose an arbitrary time-independent orthonormal basis ${\left\{\Phi_{n}\right\}}_{n=1}^{N}$ of ℋ, and let U[R]:ℋ→ℋ[R] be the unitary operator that maps ${\left\{\Phi_{n}\right\}}_{n=1}^{N}$ onto ${\left\{\phi_{n}\left[R\right]\right\}}_{n=1}^{N}$, so that
(13)$$\phi_{n}\left[R\right]=U\left[R\right]\Phi_{n}.$$This means that for all Φ∈ℋ,
(14)$$U\left[R\right]\Phi :=\sum_{n=1}^{N}\langle \Phi_{n},\Phi \rangle \;\phi_{n}\left[R\right],$$
where 〈·,·〉 stands for the inner product of ℋ.

We define a concept of covariant time derivative $D_{t}$ in ℋ by
(15)$$D_{t}\Psi \left(t\right):=\frac{d}{dt}\;\Psi \left(t\right)+i\sum_{a=1}^{d}\frac{dR_{a}\left(t\right)}{dt}\;A_{a}\left[R\left(t\right)\right]\Psi \left(t\right),$$
where *t* ranges over some interval $\left[t_{0},t_{1}\right]$ such that $R\left(t\right)\in {𝒪}_{\alpha }$, $\Psi :[t_{0},t_{1}]\to ℋ$ is a differentiable function, and for each $R\in {𝒪}_{\alpha }$ and a∈{1,2,⋯,d}, $A_{a}\left[R\right]$ is a linear operator acting in ℋ. It is the choice of these operators, or alternatively the operator-valued differential form,
$$A\left[R\right]=\sum_{a=1}^{d}dR_{a}\;A_{a}\left[R\right],$$
that determines the covariant time derivative $D_{t}\Psi \left(t\right)$ of Ψ(t). Because U[R(t)] is a unitary operator, we can use it to extend this notion of covariant time derivative to functions $\psi :\left[t_{0},t_{1}\right]\to ℋ\left[R\left(t\right)\right]$ as follows:(16)$${𝒟}_{t}\psi \left(t\right):=U\left[R\left(t\right)\right]\;D_{t}\;U{\left[R\left(t\right)\right]}^{-1}\psi \left(t\right).$$In view of (Equation 11), this is equivalent to
(17)$${𝒟}_{t}\psi \left(t\right):=\sum_{m=1}^{N}\left[\frac{dc_{m}\left(t\right)}{dt}+i\sum_{n=1}^{N}A_{mn}\left(t\right)c_{n}\left(t\right)\right]\phi_{m}\left[R\left(t\right)\right],$$
where
(18)$$\begin{array}{cc} \; & A_{mn}\left(t\right):=\sum_{a=1}^{d}\frac{dR_{a}\left(t\right)}{dt}A_{a\;mn}\left[R\left(t\right)\right], \end{array}$$
(19)$$\begin{array}{cc} \; & A_{a\;mn}\left[R\right]:=\langle \Phi_{m},A_{a}\left[R\right]\Phi_{n}\rangle . \end{array}$$

The concept of covariant differentiation defined by (Equation 16) or (Equation 17) would be acceptable provided that it does not depend on the choice of the orthonormal basis ${\left\{\phi_{n}\left[R\right]\right\}}_{n=1}^{N}$. To satisfy this requirement, the operators $A_{a}$ must comply with a particular basis transformation rule.

### 3.2. Gauge Transformations

Suppose that ${\left\{{\tilde{\phi }}_{n}\left[R\right]\right\}}_{n=1}^{N}$ is another orthonormal basis of ℋ[R] for $R\in {𝒪}_{\alpha }$. Because any two orthonormal bases of a Hilbert space are related via a unitary transformation, there is a unitary operator 𝒰[R]:ℋ[R]→ℋ[R] such that ${\tilde{\phi }}_{n}\left[R\right]=𝒰\left[R\right]\phi_{n}\left[R\right]$. Let us expand the elements ξ of ℋ[R] in both ${\left\{\phi_{n}\left[R\right]\right\}}_{n=1}^{N}$ and ${\left\{{\tilde{\phi }}_{n}\left[R\right]\right\}}_{n=1}^{N}$. This gives
(20)$$\begin{array}{cc} \; & \xi =\sum_{n=1}^{N}x_{n}\phi_{n}=\sum_{n=1}^{N}{\tilde{x}}_{n}{\tilde{\phi }}_{n}, \end{array}$$
where
(21)$$\begin{array}{cccc} \; & x_{n}:={\langle \phi_{n},\xi \rangle }_{R},\; & \; & {\tilde{x}}_{n}:={\langle {\tilde{\phi }}_{n},\xi \rangle }_{R}=\sum_{m=1}^{N}{\langle 𝒰\phi_{n},\phi_{m}\rangle }_{R}\;x_{m}, \end{array}$$
and we have suppressed the *R*-dependence of $\phi_{n}\left[R\right],{\tilde{\phi }}_{n}\left[R\right]$, and 𝒰[R] for brevity. The basis transformation ${\left\{\phi_{n}\left[R\right]\right\}}_{n=1}^{N}\to {\left\{{\tilde{\phi }}_{n}\left[R\right]\right\}}_{n=1}^{N}$ is the passive transformation,
(22)$$\begin{array}{cccc} \; & \phi_{n}\overset{𝒰}{⟶}{\tilde{\phi }}_{n}=𝒰\;\phi_{n},\; & \; & x_{n}\overset{𝒰}{⟶}{\tilde{x}}_{n}=\sum_{m=1}^{N}{\langle 𝒰\phi_{n},\phi_{m}\rangle }_{R}\;x_{m}, \end{array}$$
which does not change ξ. It induces a basis transformation in ℋ, namely
$${\left\{\Phi_{n}\right\}}_{n=1}^{N}\overset{G[R]}{⟶}{\left\{{\tilde{\Phi }}_{n}\left[R\right]\right\}}_{n=1}^{N},$$
where ${\tilde{\Phi }}_{n}\left[R\right]:=G\left[R\right]\Phi_{n}$, and G[R]:ℋ→ℋ is the unitary operator defined by
(23)$$G\left[R\right]:=U{\left[R\right]}^{-1}𝒰\left[R\right]\;U\left[R\right].$$

In Appendix B, we show that the right-hand side of (Equation 17) is left invariant under the basis transformation (Equation 22) if and only if the operators $A_{a}$ transform according to
(24)$$\begin{array}{c} A_{a}⟶{\tilde{A}}_{a}=G^{-1}A_{a}\;G-i\;G^{-1}\partial_{a}G. \end{array}$$
Here, $\partial_{a}$ stands for the partial derivative with respect to $R_{a}$, and we have suppressed the *R*-dependence of $A_{a}\left[R\right]$, ${\tilde{A}}_{a}\left[R\right]$, and G[R]. Transformation (Equation 24) coincides with a gauge transformation of the gauge field in a non-Abelian gauge theory whose gauge group is the unitary group of the Hilbert space ℋ. In differential geometry, $A_{a}\left[R\right]$ are known as the components of a local connection one form [54,55].

Suppose that $\left[t_{0},t_{1}\right]$ is a time interval such that for all $t\in [t_{0},t_{1}]$, $R\left(t\right)\in {𝒪}_{\alpha }$ and ψ(t) satisfies ${𝒟}_{t}\psi \left(t\right)=0$. Then, we say that ψ(t) is obtained by the parallel transportation of $\psi (t_{0})$ along the curve traced by R(t) in ${𝒪}_{\alpha }$. Demanding that the parallel transportation of pairs of elements of $ℋ[R\left(t_{0}\right)]$ leaves their inner product unchanged is equivalent to the condition that $A_{a}\left[R\right]$ be Hermitian operators acting in ℋ. This follows from the equivalence of ${𝒟}_{t}\psi \left(t\right)=0$ with $D_{t}\Psi \left(t\right)=0$ and the fact that we can write the latter equation as the Schrödinger equation,
(25)$$i\hbar \frac{d}{dt}\Psi \left(t\right)=H_{A}\left(t\right)\Psi \left(t\right),$$
for the Hamiltonian operator
(26)$$H_{A}\left(t\right):=\hbar \sum_{a=1}^{d}\frac{dR_{a}\left(t\right)}{dt}\;A_{a}\left[R\left(t\right)\right].$$

Because R(t) is arbitrary, the Hermiticity of the operators $A_{a}\left[R\right]$ is equivalent to the Hermiticity of the Hamiltonian operator $H_{A}\left(t\right)$ and the unitarity of the time-evolution it generates via (Equation 25). If $\psi_{1}\left(t\right)$ and $\psi_{2}\left(t\right)$ are elements of ℋ[R(t)] satisfying ${𝒟}_{t}\psi_{1}\left(t\right)={𝒟}_{t}\psi_{2}\left(t\right)=0$, the functions $\Psi_{1},\Psi_{2}:\left[t_{0},t_{1}\right]\to ℋ$ defined by
$$\begin{array}{cccc} \; & \Psi_{1}\left(t\right):=U^{-1}\left[R\left(t\right)\right]\psi_{1}\left(t\right),\; & \; & \Psi_{2}\left(t\right):=U^{-1}\left[R\left(t\right)\right]\psi_{2}\left(t\right), \end{array}$$
solve (Equation 25). Because U[R(t)] and $U{\left[R\left(t\right)\right]}^{-1}$ are unitary operators, the Hermiticity of $A_{a}\left[R\right]$, which ensures the Hermiticity of $H_{A}\left(t\right)$, implies
$$\begin{array}{ccc} {\langle \psi_{1}\left(t\right),\psi_{2}\left(t\right)\rangle }_{R(t)} & = & {\langle \psi_{1}\left(t\right),\psi_{2}\left(t\right)\rangle }_{R(t)}\\ & = & \langle U^{-1}\left[R\left(t\right)\right]\psi_{1}\left(t\right),U^{-1}\left[R\left(t\right)\right]\psi_{2}\left(t\right)\rangle \\ & = & \langle \Psi_{1}\left(t\right),\Psi_{2}\left(t\right)\rangle \\ & = & \langle \Psi_{1}\left(t_{0}\right),\Psi_{2}\left(t_{0}\right)\rangle \\ & = & {\langle \;U\left[R\left(t_{0}\right)\right]\Psi_{1}\left(t_{0}\right),U\left[R\left(t_{0}\right)\right]\Psi_{2}\left(t_{0}\right)\;\rangle }_{R(t_{0})}\\ & = & {\langle \psi_{1}\left(t_{0}\right),\psi_{2}\left(t_{0}\right)\rangle }_{R(t_{0})}. \end{array}$$

The converse of this argument also holds; ${\langle \psi_{1}\left(t\right),\psi_{2}\left(t\right)\rangle }_{R(t)}={\langle \psi_{1}\left(t_{0}\right),\psi_{2}\left(t_{0}\right)\rangle }_{R(t_{0})}$ implies $\langle \Psi_{1}\left(t\right),\Psi_{2}\left(t\right)\rangle =\langle \Psi_{1}\left(t_{0}\right),\Psi_{2}\left(t_{0}\right)\rangle$ and consequently the unitarity of the time-evolution generated by (Equation 26), which in turn requires $H_{A}\left(t\right)$ and $A_{a}\left[R\right]$ to be Hermitian operators.

### 3.3. Active Transformations of the Hilbert Space

Consider the active unitary transformations of of the Hilbert space ℋ[R]:(27)$$\xi \overset{\overset{ˇ}{𝒰}}{⟶}\overset{ˇ}{\xi }:=\overset{ˇ}{𝒰}\left[R\right]\;\xi ,$$
where $\overset{ˇ}{𝒰}\left[R\right]:ℋ\left[R\right]\to ℋ\left[R\right]$ is a unitary operator. Expanding ξ and $\overset{ˇ}{\xi }$ in the basis ${\left\{\phi_{n}\right\}}_{n=1}^{N}$ and denoting the coefficient of the expansions by $x_{n}$ and ${\overset{ˇ}{x}}_{n}$, so that $x_{n}:={\langle \phi_{n},\xi \rangle }_{R}$ and ${\overset{ˇ}{x}}_{n}:={\langle \phi_{n},\overset{ˇ}{\xi }\rangle }_{R}$, we can express (Equation 27) in the form
(28)$$\begin{array}{cccc} \; & \phi_{n}\overset{\overset{ˇ}{𝒰}}{⟶}\phi_{n},\; & \; & x_{n}\overset{\overset{ˇ}{𝒰}}{⟶}{\overset{ˇ}{x}}_{n}=\sum_{m=1}^{N}{\langle \phi_{n},\overset{ˇ}{𝒰}\phi_{m}\rangle }_{R}\;x_{m}=\sum_{m=1}^{N}{\langle {\overset{ˇ}{𝒰}}^{-1}\phi_{n},\phi_{m}\rangle }_{R}\;x_{m}. \end{array}$$

The active transformation (Equation 27) induces the following unitary transformation of the Hilbert space ℋ.
$$\Phi \overset{\overset{ˇ}{G}}{⟶}\overset{ˇ}{\Phi }:=\overset{ˇ}{G}\left[R\right]\;\Phi ,$$
where Φ∈ℋ is arbitrary and
(29)$$\overset{ˇ}{G}\left[R\right]:=U{\left[R\right]}^{-1}\overset{ˇ}{𝒰}\left[R\right]\;U\left[R\right].$$

Let us examine the effect of active unitary transformation (Equation 27) on the covariant time derivatives of the evolving state vectors ψ(t) and Ψ(t). Denoting the transformed $\psi \left(t\right),\Psi \left(t\right),{𝒟}_{t}\psi \left(t\right)$, and $D_{t}\Psi \left(t\right)$, respectively, by $\overset{ˇ}{\psi }\left(t\right),\overset{ˇ}{\Psi }\left(t\right),{\overset{ˇ}{𝒟}}_{t}\overset{ˇ}{\psi }\left(t\right)$, and ${\overset{ˇ}{D}}_{t}\overset{ˇ}{\Psi }\left(t\right)$, we have
(30)$$\begin{array}{cccc} \; & \overset{ˇ}{\psi }\left(t\right)=\overset{ˇ}{𝒰}\left[R\left(t\right)\right]\psi \left(t\right),\; & \; & \overset{ˇ}{\Psi }\left(t\right)=\overset{ˇ}{G}\left[R\left(t\right)\right]\Psi \left(t\right), \end{array}$$
and
$$\begin{array}{cc} {\overset{ˇ}{𝒟}}_{t}\overset{ˇ}{\psi }\; & =U\;{\overset{ˇ}{D}}_{t}\;U^{-1}\tilde{\psi }=U\;{\overset{ˇ}{D}}_{t}\overset{ˇ}{\Psi }\\ \; & =U\left(\frac{d}{dt}\overset{ˇ}{\Psi }+i\sum_{a=1}^{d}\frac{dR_{a}}{dt}\;{\overset{ˇ}{A}}_{a}\overset{ˇ}{\Psi }\right),\\ \; & =U\left(\overset{ˇ}{G}\;\frac{d}{dt}\Psi +\frac{d\overset{ˇ}{G}}{dt}\Psi +i\sum_{a=1}^{d}\frac{dR_{a}}{dt}\;{\overset{ˇ}{A}}_{a}\overset{ˇ}{G}\;\Psi \right)\\ \; & =U\;\overset{ˇ}{G}\left[D_{t}\Psi +{\overset{ˇ}{G}}^{-1}\frac{d\overset{ˇ}{G}}{dt}\Psi +i\sum_{a=1}^{N}\frac{dR_{a}}{dt}\left({\overset{ˇ}{G}}^{-1}{\overset{ˇ}{A}}_{a}\;\overset{ˇ}{G}-A_{a}\right)\Psi \right]\\ \; & =\overset{ˇ}{𝒰}\;U\;D_{t}\Psi +i\;\overset{ˇ}{𝒰}\;U\sum_{a=1}^{N}\frac{dR_{a}}{dt}\left({\overset{ˇ}{G}}^{-1}{\overset{ˇ}{A}}_{a}\;\overset{ˇ}{G}-A_{a}-i\;{\overset{ˇ}{G}}^{-1}\partial_{a}\overset{ˇ}{G}\right)\Psi \\ \; & =\overset{ˇ}{𝒰}\;{𝒟}_{t}\psi +i\;\overset{ˇ}{𝒰}\;U\;{\overset{ˇ}{G}}^{-1}\sum_{a=1}^{N}\frac{dR_{a}}{dt}\left[{\overset{ˇ}{A}}_{a}-\overset{ˇ}{G}\;A_{a}{\overset{ˇ}{G}}^{-1}-i\left(\partial_{a}\overset{ˇ}{G}\right)\;{\overset{ˇ}{G}}^{-1}\right]\overset{ˇ}{G}\;\Psi \\ \; & =\overset{ˇ}{𝒰}\;{𝒟}_{t}\psi +i\;U\sum_{a=1}^{N}\frac{dR_{a}}{dt}\left[{\overset{ˇ}{A}}_{a}-\overset{ˇ}{G}\;A_{a}{\overset{ˇ}{G}}^{-1}-i\left(\partial_{a}\overset{ˇ}{G}\right)\;{\overset{ˇ}{G}}^{-1}\right]U^{-1}\overset{ˇ}{\psi }, \end{array}$$
where we have used (Equation 16), (Equation 17), (Equation 29), and (Equation 30). This calculation shows that under the active transformation (Equation 27), ${𝒟}_{t}\psi$ transforms according to
(31)$${𝒟}_{t}\psi \overset{\overset{ˇ}{𝒰}}{⟶}{\overset{ˇ}{𝒟}}_{t}\overset{ˇ}{\psi }=\overset{ˇ}{𝒰}\;{𝒟}_{t}\psi ,$$
if and only if
(32)$$A_{a}\overset{\overset{ˇ}{𝒰}}{⟶}{\overset{ˇ}{A}}_{a}=\overset{ˇ}{G}\;A_{a}{\overset{ˇ}{G}}^{-1}+i\left(\partial_{a}\overset{ˇ}{G}\right)\;{\overset{ˇ}{G}}^{-1}.$$

If we demand that ${𝒟}_{t}\psi \left(t\right)$ belongs to ℋ[R(t)], then under active unitary transformations of ℋ[R(t)], it must transform similarly to ψ(t), i.e., (Equation 31) and consequently (Equation 32) hold. It is also easy to see that (Equation 31) is equivalent to
(33)$${𝒟}_{t}\overset{\overset{ˇ}{𝒰}}{⟶}{\overset{ˇ}{𝒟}}_{t}=\overset{ˇ}{𝒰}\;{𝒟}_{t}\;{\overset{ˇ}{𝒰}}^{-1}.$$This is the main reason for calling ${𝒟}_{t}$ the “covariant differentiation with respect to time.” We can view (Equation 32) as an implication of (Equation 33).

### 3.4. Covariant Schrödinger Equation

Having introduced the notion of covariant time derivative and examined some of its basic properties, we propose to determine the dynamics of our quantum system in ℋ(t) using the following covariant generalization of the Schrödinger equation.
(34)$$i\hbar {𝒟}_{t}\psi \left(t\right)=H\left(t\right)\psi \left(t\right),$$
where H(t):=H[R(t)], and H[R] is a Hermitian operator acting in ℋ[R] for all $R\in {𝒪}_{\alpha }$. This provides a local description of the dynamics of the system in the sense that the curve traced by R(t) is confined to a single open subset ${𝒪}_{\alpha }$ of *M*. We can extend this prescription to situations where R(t) traces an arbitrary smooth curve, if we know the structure functions of the underlying Hermitian vector bundle E, [40].

We can use the dynamics generated by (Equation 34) in ℋ[R(t)] and the unitary operator $U^{-1}\left[R\left(t\right)\right]$ to induce a dynamics in the time-independent Hilbert space ℋ. Applying $U^{-1}\left[R\left(t\right)\right]$ to both sides of (Equation 34) and using (Equation 15) and (Equation 16), we arrive at the Schrödinger equation
(35)$$i\hbar \frac{d}{dt}\Psi \left(t\right)=H\left(t\right)\Psi \left(t\right),$$
where
(36)$$\begin{array}{cccccc} \; & H\left(t\right):=H_{A}\left(t\right)+H_{E}\left(t\right),\; & \; & H_{E}\left(t\right):=H_{E}\left[R\left(t\right)\right],\; & \; & H_{E}\left[R\right]:=U{\left[R\right]}^{-1}H\left[R\right]\;U\left[R\right], \end{array}$$
and $H_{A}\left(t\right)$ is the operator defined by (Equation 26). Under the active unitary transformations $\overset{ˇ}{𝒰}\left[R\left(t\right)\right]$ of the Hilbert space ℋ(t), the operators H(t), $H_{A}\left(t\right)$, and $H_{E}\left(t\right)$ transform according to
(37)$$\begin{array}{cc} \; & H\overset{\overset{ˇ}{𝒰}}{⟶}\overset{ˇ}{H}=\overset{ˇ}{𝒰}\;H\;{\overset{ˇ}{𝒰}}^{-1}, \end{array}$$
(38)$$\begin{array}{cc} \; & H_{A}\overset{\overset{ˇ}{𝒰}}{⟶}{\overset{ˇ}{H}}_{A}=\overset{ˇ}{G}\;H_{A}{\overset{ˇ}{G}}^{-1}+i\hbar \;\frac{d\overset{ˇ}{G}}{dt}\;{\overset{ˇ}{G}}^{-1}, \end{array}$$
(39)$$\begin{array}{cc} \; & H_{E}\overset{\overset{ˇ}{𝒰}}{⟶}{\overset{ˇ}{H}}_{E}=\overset{ˇ}{G}\;H_{E}\;{\overset{ˇ}{G}}^{-1}, \end{array}$$Relation (Equation 37) follows from (Equation 33) and (Equation 35), while (Equation 38) and (Equation 39) are consequences of (Equation 26), (Equation 32), (Equation 36), and (Equation 37). In view of the first equation in (Equation 36) and (Equation 37)–(Equation 39), the Hamiltonian H(t) transforms according to
(40)$$H\overset{\overset{ˇ}{𝒰}}{⟶}\overset{ˇ}{H}=\overset{ˇ}{G}\;H\;{\overset{ˇ}{G}}^{-1}+i\hbar \;\frac{d\overset{ˇ}{G}}{dt}\;{\overset{ˇ}{G}}^{-1}.$$

For situations where R(t) traces a curve contained in ${𝒪}_{\alpha }$, we can describe our quantum system using either of the Hilbert space–Hamiltonian pairs (ℋ(t),H(t)) and (ℋ,H(t)). Let us first examine its description in terms of (ℋ,H(t)), where the Hilbert space is time-independent. As we see from the above derivation of (Equation 39), the last term on the right-hand side of this equation has its origin in the way $H_{A}\left(t\right)$, and consequently the local connection components $A_{a}\left[R\right]$, transform under active unitary transformations of the Hilbert space ℋ. The latter do not correspond to observables of the system. They describe the geometry of the bundle E of Hilbert spaces ℋ[R]. In contrast, $H_{E}\left(t\right)$ is a Hermitian operator, which we can identify with an observable of the system. We propose to interpret it as the energy observable. Because H(t) is the counterpart of $H_{E}\left(t\right)$ in the representation of our system in terms of (ℋ(t),H(t)), the same interpretation applies to H(t). The advantage of the latter representation is that it applies to all of *M*, i.e., it provides a global description of the quantum system that is applicable regardless of the choice of the curve traced by the parameters R(t) of the system in time.

If we confine our attention to a single curve C of parameters, then the bundle over this curve is topologically trivial and there is no advantage of using (ℋ(t),H(t)). Yet the existence of this representation and the corresponding covariant Schrödinger equation reveal the basic reason why the Hamiltonian transforms differently from the observables under time-dependent unitary transforms of the Hilbert space. This follows from a hidden geometric aspect of quantum mechanics, which is quantified in terms of a Hermitian operator-valued gauge field A[R].

When we describe a quantum system using a time-independent Hilbert space, we neglect the presence of A[R]. This does not cause any problems, if A[R] is a pure gauge (a flat connection). In this case, we can make a proper choice for the gauge (basis) where A[R]=0. In other words, we assume the existence of a representation of the system in terms of a Hilbert space–Hamiltonian pair with a time-independent Hilbert space in which $A_{a}\left[R\left(t\right)\right]=0$. The developments reported in Refs. [40,41] and the present article question the validity of this assumption. They suggest that we should investigate the physical content of quantum systems for which the connection one-form A[R] fails to be flat and explore the physical meaning of the corresponding local curvature two form,
(41)F[R]:=dA[R]+iA[R]∧A[R],
which is a measure of the non-flatness of A[R], [54,55]. We can express F[R] in terms of its components $F_{ab}\left[R\right]$ according to
(42)$$F\left[R\right]=\frac{1}{2}\sum_{a,b=1}^{d}dR_{a}\wedge dR_{b}\;F_{ab}\left[R\right],$$
where ∧ stands for the standard wedge product of differential forms [54,55]. In view of (Equation 41) and (Equation 42), $F_{ab}\left[R\right]$ are Hermitian operators acting in ℋ[R] that are given by
$$F_{ab}=\partial_{a}A_{b}-\partial_{b}A_{a}+i\left[A_{a},A_{b}\right].$$These correspond to the components of the field strength associated with the gauge field $A_{a}$.

### 3.5. Energy Observable and Reparametrizations of Time

In the preceding subsections we consider non-stationary quantum systems described by time-dependent Hilbert space–Hamiltonian pairs where the time-dependence of the Hilbert space and the Hamiltonian is determined through the dependence of a set of real parameters $R=(R_{1},R_{2},\cdots ,R_{d})$ on time. These trace a curve C in a parameter space *M*, and a consistent description of the dynamics of the system is achieved by replacing the role of the Hilbert space by a bundle $E_{C}$ of Hilbert spaces over C. This is obtained as the restriction of a Hermitian vector bundle E over *M* to C, which is necessarily a trivial Hermitian vector bundle. This does not, however, mean that $E_{C}$ has a trivial geometry (a flat connection).

The dynamics of the system is determined by the covariant Schrödinger Equation (Equation 34), which involves Hermitian operators H[R(t)] acting in the fibers ℋ[R(t)] of $E_{C}$. These are given by the restriction of H[R] to C. We can view H as a function that maps *M* to a real vector bundle u over *M*. The fibers $u_{R}$ of u that are attached to the points R∈M are vector spaces of Hermitian operators acting in ℋ[R]. Because H[R] acts in ℋ[R], H:M→u is a function such that $H\left[R\right]\in u_{R}$ for all R∈M. This identifies H with what mathematicians call a global section of u, [40,41]. The restriction of H to C specifies the Hermitian operators H[R(t)], which we identify with the energy observable. Other observables of the system are also obtained similarly by restricting global sections of u to C, [40].

Now, consider a reparameterization, t→τ:=f(t), of C where $f:[t_{0},t_{1}]\to R$ is a monotonically decreasing or increasing differentiable function. Such a function determines a reparameterization of C provided that R(t) and $\overset{˘}{R}\left(t\right):=R\left(\tau \right)=R\left(f\left(t\right)\right)$ trace the same segment of C. Under the reparameterization, t→τ, $\frac{dR_{a}\left(t\right)}{dt}$ and consequently ${𝒟}_{t}$ transform according to
(43)$$\begin{array}{cccc} \; & \frac{dR_{a}\left(t\right)}{dt}⟶\frac{d{\overset{˘}{R}}_{a}\left(t\right)}{dt}=\frac{1}{g(\tau )}\frac{dR_{a}\left(\tau \right)}{d\tau },\; & \; & {𝒟}_{t}⟶{\overset{˘}{𝒟}}_{t}=\frac{1}{\tau^{\prime }\left(t\right)}{𝒟}_{\tau (t)}=\frac{1}{g(\tau )}{𝒟}_{\tau }, \end{array}$$
where
$$g\left(\tau \right):=f^{\prime }\left(f^{-1}\left(\tau \right)\right)=f^{\prime }\left(t\right){|}_{t=f^{-1}\left(\tau \right)},$$$f^{\prime }$ denotes the derivative of *f*, and we have made use of (Equation 17). In view of (Equation 43), the reparameterization, t→τ, changes the covariant Schrödinger Equation (Equation 34) into
(44)$$i\hbar {𝒟}_{\tau }\psi \left(\tau \right)=g\left(\tau \right)H\left(\tau \right)\psi \left(\tau \right),$$
where H(τ)=H[R(τ)]. Comparing (Equation 34) and (Equation 44), we see that whenever H=0, the solutions ψ(t) of the covariant Schrödinger Equation (Equation 34) are invariant under reparameterizations, t→τ. This is to be expected, because in this case ψ(t) is obtained via parallel transportation of $\psi (t_{0})$ along the curve C. Because we interpret *t* as time, we say that the dynamics of the system is time-reparameterization-invariant.

The left-hand side of the covariant Schrödinger Equation (Equation 34) involves the covariant time derivative determined by the connection on E, a quantity which specifies the geometry of E, while its right-hand side is given by H[R(t)], which is the energy observable at time *t*. The latter represents the interactions that are independent of the geometry of E. The above analysis shows that these interactions are responsible for the breakdown of the time-reparameterization invariance of the dynamics of system.

### 3.6. Equivalent Representations of Quantum Systems

A by-product of our geometric treatment of quantum systems described by time-dependent Hilbert space–Hamiltonian pairs (ℋ(t),H(t)) is that they admit a consistent formulation in terms of time-independent Hilbert space–Hamiltonian pairs (ℋ,H(t)). However, the choice of the latter is not unique. Two Hilbert space–Hamiltonian pairs, $\left({ℋ}_{1},H_{1}\left(t\right)\right)$ and $\left({ℋ}_{2},H_{2}\left(t\right)\right)$, represent the same quantum system if there is a one-to-one correspondence between state vectors and Hermitian operators of ${ℋ}_{1}$ and those of ${ℋ}_{2}$ such that the expectation values of the corresponding Hermitian operators in the corresponding state vectors coincide, and the solutions of the Schrödinger equation defined by $H_{1}\left(t\right)$ in ${ℋ}_{1}$ correspond to those of the Schrödinger equation defined by $H_{2}\left(t\right)$ in ${ℋ}_{2}$. Such a correspondence defines a possibly time-dependent everywhere-defined one-to-one and onto a linear operator $U\left(t\right):{ℋ}_{1}\to {ℋ}_{2}$ with the following properties.

(P1)For all t∈R, $\xi_{1}\in {ℋ}_{1}$, and Hermitian operator $O_{1}:{ℋ}_{1}\to {ℋ}_{2}$ other than $H_{1}\left(t\right)$, there are a unique vector $\xi_{2}\in {ℋ}_{2}$ and a unique Hermitian operator $O_{2}:{ℋ}_{1}\to {ℋ}_{2}$ such that
(45)$$\begin{array}{cccc} \; & \xi_{2}=U\left(t\right)\xi_{1},\; & \; & O_{2}=U\left(t\right)O_{1}U{\left(t\right)}^{-1}. \end{array}$$This determines the corresponding state vectors and Hermitian operators of ${ℋ}_{1}$ and ${ℋ}_{2}$.(P2)Let t∈R be arbitrary and ${\langle \cdot ,\cdot \rangle }_{1}$ and ${\langle \cdot ,\cdot \rangle }_{2}$ denote the inner products of ${ℋ}_{1}$ and ${ℋ}_{2}$, respectively. Then,
(46)$${\langle \xi_{1},\zeta_{1}\rangle }_{1}={\langle \;U\left(t\right)\xi_{1},U\left(t\right)\zeta_{1}\rangle }_{2}\;\;\;\mathrm{for}\;\mathrm{all}\;\;\;\xi ,\zeta_{1}\in {ℋ}_{1}.$$This means that U(t) is a unitary operator [43,45].(P3)For all t∈R,
(47)$$i\hbar \frac{d}{dt}U\left(t\right)=H_{2}\left(t\right)U\left(t\right)-U\left(t\right)H_{1}\left(t\right).$$

Now, suppose that a state and an observable of the quantum system are, respectively, given by the state vector $\xi_{1}$ and Hermitian operator $O_{1}$ in its $\left({ℋ}_{1},H_{1}\left(t\right)\right)$ representation, and $\psi_{1}\left(t\right)$ is an evolving state vector in this representation, i.e., it solves
(48)$$i\hbar \frac{d}{dt}\psi_{1}\left(t\right)=H_{1}\left(t\right)\psi_{1}\left(t\right).$$
Then, according to P1 the corresponding state vector, Hermitian operator, and evolving state vector in the $\left({ℋ}_{2},H_{2}\left(t\right)\right)$ representation are given by (Equation 45) and $\psi_{2}\left(t\right)=U\left(t\right)\psi_{1}\left(t\right)$. In view of these equations, P2, P3, and (Equation 48), we have
$$\begin{array}{cccc} \; & \frac{{\langle \xi_{1},O_{1}\xi_{1}\rangle }_{1}}{{\langle \xi_{1},\xi_{1}\rangle }_{1}}=\frac{{\langle \xi_{2},O_{2}\xi_{2}\rangle }_{2}}{{\langle \xi_{2},\xi_{2}\rangle }_{2}},\; & \; & i\hbar \frac{d}{dt}\psi_{2}\left(t\right)=H_{2}\left(t\right)\psi_{2}\left(t\right). \end{array}$$This shows that we can quantify the kinematic and dynamical properties of the system using either of $\left({ℋ}_{1},H_{1}\left(t\right)\right)$ and $\left({ℋ}_{2},H_{2}\left(t\right)\right)$, i.e., they represent the same quantum system.

Next, we address the question of whether two Hilbert space–Hamiltonian pairs, $\left({ℋ}_{1},H_{1}\left(t\right)\right)$ and $\left({ℋ}_{2},H_{2}\left(t\right)\right)$, represent the same quantum system. The above analysis shows that this is the case if and only if there is a unitary operator $U\left(t\right):{ℋ}_{1}\to {ℋ}_{2}$ fulfilling (Equation 47). Because equality of the dimensions of any two (separable) Hilbert spaces is equivalent to the existence of a unitary operator mapping one to the other [43], a necessary condition for $\left({ℋ}_{1},H_{1}\left(t\right)\right)$ and $\left({ℋ}_{2},H_{2}\left(t\right)\right)$ to represent the same quantum system is that ${ℋ}_{1}$ and ${ℋ}_{2}$ have the same dimension. This is, however, not a sufficient condition; if the dimensions of ${ℋ}_{1}$ and ${ℋ}_{2}$ coincide, there are always unitary operators $U\left(t\right):{ℋ}_{1}\to {ℋ}_{2}$, but they may violate (Equation 47).

## 4. Time-Dependent Inner Products

Given a complex vector space *V* and an inner product 〈·|·〉 on *V*, every inner product ≺·,·≻ on this vector space has the form
(49)$$≺\cdot ,\cdot ≻=\langle \cdot |\eta_{+}\cdot \rangle ,$$
where $\eta_{+}:V\to V$ is a linear operator defined on all of *V* such that
(50)$$\langle \xi |\eta_{+}\xi \rangle \in R^{+}\;\;\;\mathrm{for}\;\;\;\xi \in V\setminus \left\{o\right\},$$
and *o* is the zero vector in *V*. Let us make the $\eta_{+}$-dependence of ≺·,·≻ transparent by labeling the latter by ${\langle \cdot ,\cdot \rangle }_{\eta_{+}}$, i.e., set
(51)$${\langle \cdot ,\cdot \rangle }_{\eta_{+}}:=≺\cdot ,\cdot ≻,$$
use ℋ and ${ℋ}_{\eta_{+}}$ to denote the inner product spaces obtained by endowing *V* with the inner products 〈·|·〉 and ${\langle \cdot ,\cdot \rangle }_{\eta_{+}}$, respectively, i.e.,
$$\begin{array}{cccc} \; & ℋ:=(V,\langle \cdot |\cdot \rangle ),\; & \; & {ℋ}_{\eta_{+}}:=\left(V,{\langle \cdot ,\cdot \rangle }_{\eta_{+}}\right), \end{array}$$
and suppose that ℋ and ${ℋ}_{\eta_{+}}$ are Hilbert spaces. Because $\eta_{+}$ determines the inner product ${\langle \cdot ,\cdot \rangle }_{\eta_{+}}$, it is called a metric operator on ℋ.

A linear operator L:ℋ→ℋ with domain 𝒟 is said to be continuous, if for every sequence ${\left\{\xi_{n}\right\}}_{n=1}^{\infty }$ in 𝒟 that converges to some ξ∈𝒟, the sequence ${\left\{L\xi_{n}\right\}}_{n=1}^{\infty }$ converges to Lξ. *L* is said to be bounded, if there is some $\alpha \in R^{+}$ such that for all ς∈𝒟, ‖Lς‖≤α‖ς‖. It is easy to show that *L* is continuous if and only if it is bounded.

Condition (Equation 50) identifies $\eta_{+}$ with a positive-definite operator $\eta_{+}:ℋ\to ℋ$ defined on all of ℋ. This in particular implies that it is an everywhere-defined Hermitian operator with an everywhere-defined Hermitian inverse $\eta_{+}^{-1}:ℋ\to ℋ$, [21]. A basic theorem of operator theory known as Hellinger–Toeplitz theorem states that everywhere-defined Hermitian operators cannot be discontinous. This implies that both $\eta_{+}$ and $\eta_{+}^{-1}$ are continuous and consequently bounded operators. For this reason, ℋ and ${ℋ}_{\eta_{+}}$ have identical topological properties, i.e., the notions of open subset, convergence, limit, and continuity in ℋ and ${ℋ}_{\eta_{+}}$ coincide.

If in (Equation 49) we replace $\eta_{+}$ by a Hermitian automorphism, η:ℋ→ℋ, i.e., a Hermitian operator that is defined on all of ℋ and is one-to-one and onto, ≺·,·≻ defines a possibly indefinite inner product on ℋ, [56]. For this reason we call η a pseudo-metric operator [21].

Now, consider a quantum system represented by the Hilbert space–Hamiltonian pair (ℋ,H(t)) where H(t):ℋ→ℋ is a possibly non-Hermitian operator with a dense domain, i.e., for every ξ∈ℋ, there is a sequence in Dom(H) that converges to ξ. Suppose that for all $t_{0}\in R$ and $\psi_{0}\in ℋ$, the Schrödinger equation,
(52)$$i\hbar \frac{d}{dt}\psi \left(t\right)=H\left(t\right)\psi \left(t\right),$$
has a global solution (defined for all t∈R) fulfilling the initial condition, $\psi \left(t_{0}\right)=\psi_{0}$. The time evolution determined by (Equation 52) in ℋ is necessarily non-unitary, i.e., 〈ψ(t),ψ(t)〉 depends on time. It might, however, be possible to find a metric operator $\eta_{+}\left(t\right)$ such that the inner product ${\langle \cdot ,\cdot \rangle }_{\eta_{+}\left(t\right)}$ is invariant under the time evolution, i.e., for every pair, $\psi_{1}\left(t\right)$ and $\psi_{2}\left(t\right)$, of the solutions of (Equation 52), ${\langle \psi_{1},\psi_{2}\rangle }_{\eta_{+}\left(t\right)}$ is time-independent. In view of (Equation 49), this condition is equivalent to
(53)$$\frac{d}{dt}\langle \psi_{1}\left(t\right)|\eta_{+}\left(t\right)\psi_{2}\left(t\right)\rangle =0.$$

Using (Equation 52) and arbitrariness of the solutions, $\psi_{1}\left(t\right)$ and $\psi_{2}\left(t\right)$, we can express (Equation 53) in the form of the following differential equation for $\eta_{+}\left(t\right)$, [31].
(54)$$i\hbar \frac{d}{dt}\eta_{+}\left(t\right)=\eta_{+}\left(t\right)H\left(t\right)-H{\left(t\right)}^{†}\eta_{+}\left(t\right),$$
where $\frac{d}{dt}\eta_{+}\left(t\right)$ stands for the strong derivative of $\eta_{+}\left(t\right)$, i.e., it is the operator that satisfies
$$\frac{d}{dt}\eta_{+}\left(t\right)\;\phi =\lim_{\epsilon \to 0}\frac{1}{\epsilon }\left[\eta_{+}\left(t+\epsilon \right)\phi -\eta_{+}\left(t\right)\phi \right]\;\;\;\mathrm{for}\;\mathrm{all}\;\;\;\phi \in ℋ.$$If H(t) is Hermitian, (Equation 54) reduces to the Liouville–von Neumann equation.

It is easy to infer from (Equation 53) that the general solution of (Equation 53) has the form
(55)$$\eta_{+}\left(t\right)=U{\left(t,t_{0}\right)}^{-1†}\eta_{0}\;U{\left(t,t_{0}\right)}^{-1},$$
where $t_{0}\in R$, $U(t,t_{0})$ is the evolution operator associated with the Hamiltonian operator H(t), and $\eta_{0}$ is a time-independent metric operator [23]. Clearly,
(56)$$\eta_{+}\left(t_{0}\right)=\eta_{0}.$$We also recall that $U(t,t_{0})$ satisfies
(57)$$\begin{array}{cccc} \; & i\hbar \frac{d}{dt}U\left(t,t_{0}\right)=H\left(t\right)\;U\left(t,t_{0}\right),\; & \; & U(t_{0},t_{0})=I, \end{array}$$
where $\frac{d}{dt}U\left(t,t_{0}\right)$ is the strong derivative of $U(t,t_{0})$ with respect to *t* in ℋ, and *I* is the identity operator.

As a linear operator mapping ℋ onto ℋ, $U(t,t_{0})$ fails to be a unitary operator unless if $\eta_{+}\left(t\right)=I$ for all t∈R. According to (Equation 54), this happens when H(t) is Hermitian. As an operator mapping ${ℋ}_{\eta_{+}\left(t_{0}\right)}$ to ${ℋ}_{\eta_{+}\left(t\right)}$, however, $U(t,t_{0})$ is a unitary operator. This is because for all $t_{0},t\in R$ and $\xi ,\zeta \in {ℋ}_{\eta_{+}\left(t_{0}\right)}$,
$$\begin{array}{cc} {\langle \;U\left(t,t_{0}\right)\xi \;,\;U\left(t,t_{0}\right)\zeta \;\rangle }_{\eta_{+}\left(t\right)} & =\langle \;U\left(t,t_{0}\right)\xi \;|\;\eta_{+}\left(t\right)\;U\left(t,t_{0}\right)\zeta \;\rangle \\ \; & =\langle \xi |\;U{\left(t,t_{0}\right)}^{†}\eta_{+}\left(t\right)\;U\left(t,t_{0}\right)\zeta \;\rangle \\ \; & =\langle \xi |\eta_{0}\zeta \rangle \\ \; & =\langle \xi |\eta_{+}\left(t_{0}\right)\zeta \rangle \\ \; & ={\langle \xi |\zeta \rangle }_{\eta_{+}\left(t_{0}\right)}, \end{array}$$
where we have made use of (Equation 49), (Equation 51), (Equation 55) and (Equation 56).

Next, consider the positive square root of $\eta_{+}\left(t\right)$, which we denote by ρ(t). This is the unique positive-definite operator acting in ℋ that satisfies $\rho {\left(t\right)}^{2}=\eta_{+}\left(t\right)$, [43]. In view of (Equation 49) and (Equation 51), for all $\phi ,\chi \in {ℋ}_{\eta_{+}\left(t\right)}$,
$$\begin{array}{c} {\langle \phi ,\chi \rangle }_{\eta_{+}\left(t\right)}=\langle \phi |\eta_{+}\left(t\right)\chi \rangle =\langle \phi |\rho {\left(t\right)}^{2}\chi \rangle =\langle \rho \left(t\right)\phi |\rho \left(t\right)\chi \rangle . \end{array}$$This calculation shows that as an operator mapping ${ℋ}_{\eta_{+}\left(t_{0}\right)}$ to ℋ, ρ is a unitary operator [16]. Consequently, given a densely defined linear operator $O\left(t\right):{ℋ}_{\eta_{+}\left(t_{0}\right)}\to {ℋ}_{\eta_{+}\left(t_{0}\right)}$ and
$$o\left(t\right):=\rho \left(t\right)O\left(t\right)\rho {\left(t\right)}^{-1},$$O(t) is a Hermitian operator acting in ${ℋ}_{\eta_{+}\left(t_{0}\right)}$ if and only if o(t) is a Hermitian operator acting in ℋ. Expressing O(t) in terms of o(t), using the result to compute its adjoint in ℋ, and assuming that o(t) is Hermitian, we have
(58)$$\begin{array}{cc} O{\left(t\right)}^{†} & ={\left[\rho {\left(t\right)}^{-1}o\left(t\right)\rho \left(t\right)\right]}^{†}=\rho \left(t\right)o\left(t\right)\rho {\left(t\right)}^{-1}\\ \; & =\rho {\left(t\right)}^{2}O\left(t\right)\rho {\left(t\right)}^{-2}=\eta_{+}\left(t\right)O\left(t\right)\eta_{+}{\left(t\right)}^{-1}. \end{array}$$This calculation identifies Hermitian operators O(t) acting in ${ℋ}_{\eta_{+}\left(t_{0}\right)}$ with $\eta_{+}\left(t\right)$-pseudo-Hermitian operators acting in ℋ, [13,14,16,17,21].

Solving (Equation 54) for $H{\left(t\right)}^{†}$, we find
(59)$$H{\left(t\right)}^{†}=\eta_{+}\left(t\right)H\left(t\right)\eta_{+}{\left(t\right)}^{-1}+i\hbar \left[\frac{d}{dt}\eta \left(t\right)\right]\eta_{+}{\left(t\right)}^{-1}.$$
Comparing this equation with (Equation 58), we see that H(t) fails to be a Hermitian operator acting in ${ℋ}_{\eta_{+}\left(t\right)}$ unless $\eta_{+}\left(t\right)$ is time-independent, i.e., $\eta_{+}\left(t\right)=\eta_{0}$ for all t∈R.

If we identify ${ℋ}_{\eta_{+}\left(t\right)}$ with the Hilbert space of our quantum system, the time evolution generated by the Hamiltonian H(t) becomes unitary provided that we solve the Schrödinger Equation (Equation 52) in ℋ but compute the transition probabilities and the expectation values of the observables as if the state vectors of the system belong to ${ℋ}_{\eta_{+}\left(t\right)}$ and the observables are Hermitian operators acting in ${ℋ}_{\eta_{+}\left(t\right)}$. This leads us to the unavoidable conclusion that the non-Hermitian Hamiltonian operator H(t), which determines the time evolution in ℋ, does not act as a Hermitian operator in ${ℋ}_{\eta_{+}\left(t\right)}$. Therefore, it cannot represent an observable of the quantum system [32]. The geometric formulation of dynamics of the system that we described in Section 3 shows that this is a generic feature of all quantum systems.

The non-Hermitian Hamiltonian operator H(t) is not a Hermitian operator acting in ${ℋ}_{\eta_{+}\left(t\right)}$, but the time-dependent unitary operator $\rho \left(t\right):{ℋ}_{\eta_{+}\left(t\right)}\to ℋ$ maps it to a Hermitian operator acting in ℋ, namely,
(60)$$h\left(t\right):=\rho \left(t\right)H\left(t\right)\rho {\left(t\right)}^{-1}+i\hbar \;\frac{d\rho (t)}{dt}\;\rho {\left(t\right)}^{-1}.$$For this reason, we can represent the quantum system using the Hilbert space–Hamiltonian pair (ℋ,h(t)), [17,21].

The above developments rely on the use of ρ(t), which is the positive square root of η(t) in ℋ. One can instead consider a more general decomposition of η(t) in terms of a possibly non-Hermitian operator A(t):ℋ→ℋ, which satisfies
(61)$$\eta \left(t\right)=A{\left(t\right)}^{†}A\left(t\right),$$
and as a linear operator mapping ${ℋ}_{\eta_{+}\left(t\right)}$ to ℋ is unitary. For situations where H has a discrete spectrum and its eigenvectors form a Riesz basis [21] of ℋ, one can identify A(t) with the linear operator that maps this basis onto an orthonormal basis of ℋ, [14].

Because ρ(t) is also a unitary operator mapping ${ℋ}_{\eta_{+}\left(t\right)}$ to ℋ, $u\left(t\right):=A\left(t\right)\rho {\left(t\right)}^{-1}$ is a unitary operator mapping ℋ onto ℋ. Clearly,
(62)A(t)=u(t)ρ(t).This equation shows that given a metric operator η(t) fulfilling (Equation 59), every choice for the operator A(t) that achieves the decomposition (Equation 61) is uniquely determined by a possibly time-dependent unitary operator u(t):ℋ→ℋ.

We can use A(t) to obtain an equivalent representation of the quantum system, namely $\left(ℋ,h_{A}\left(t\right)\right)$, where $h_{A}\left(t\right)$ is the Hamiltonian operator defined in ℋ by
$$\begin{array}{cc} h_{A}\left(t\right) & :=A\left(t\right)H\left(t\right)A{\left(t\right)}^{-1}+i\hbar \;\frac{dA(t)}{dt}\;A{\left(t\right)}^{-1}\\ \; & =u\left(t\right)\left[\rho \left(t\right)H\left(t\right)\rho {\left(t\right)}^{-1}+i\hbar \;\frac{d\rho (t)}{dt}\;\rho {\left(t\right)}^{-1}\right]u{\left(t\right)}^{-1}+i\hbar \;\frac{du(t)}{dt}\;u{\left(t\right)}^{-1}\\ \; & =u\left(t\right)h\left(t\right)u{\left(t\right)}^{-1}+i\hbar \;\frac{du(t)}{dt}\;u{\left(t\right)}^{-1}. \end{array}$$It is easy to see that $h_{A}\left(t\right)$ is a Hermitian operator acting in ℋ. This calculation shows that the Hermitization of H(t) that makes use of ρ(t) is unique up to unitary equivalence, i.e., every representation of the quantum system that has ℋ as its Hilbert space is obtained from (ℋ,h(t)) via a unitary transformation u(t) of ℋ.

We close this section by making the following comments:There is a major difference between the consequences of (Equation 54) for time-dependent and time-independent metric operators. As we mentioned above, we can solve this equation for $\eta_{+}\left(t\right)$ for every Hamiltonian H(t) that admits an evolution operator $U(t,t_{0})$. For example, when ℋ is finite-dimensional or more generally H(t) is a bounded operator, $U(t,t_{0})$ exists [43], and (Equation 55) provides the general solution of (Equation 54). Therefore (Equation 54) does not put any restriction on the choice of the Hamiltonian operator H(t) or its spectrum. The situation is drastically different if we demand the existence of a time-independent metric operator. In this case, (Equation 54) reduces to the pseudo-Hermiticity condition for H(t), which imposes severe restrictions on its spectrum [13,14,15]. In this sense, referring to (Equation 54) as time-dependent pseudo-Hermiticity or a quasi-Hermiticity condition is misleading, for it does not impose any condition on the Hamiltonian.The results of this section are related to the constructions outlined in Section 3. If we denote the isomorphisms that map the fibers ℋ[R] of the Hermitian vector bundle E over ${𝒪}_{\alpha }$ to its typical fiber ℋ by $\phi_{\alpha }\left[R\right]:ℋ\left[R\right]\to ℋ$, as we describe in Ref. [40], we can use them together with the inner product of ℋ[R] to induce an inner product on ℋ. This leads to an example of the Hilbert space ${ℋ}_{\eta_{+}}$ with $\eta_{+}$ depending on *R*. We can then identify the unitary operator U[R] of Section 4 with ${\left(\rho \left[R\right]\phi_{\alpha }\left[R\right]\right)}^{-1}$ where ρ[R] is the positive square root of $\eta_{+}\left[R\right]$.

## 5. Concluding Remarks

There are many examples of physical interest where a quantum system is described by a time-dependent Hilbert space. Dealing with such a system requires extra care, because the very notion of the time derivative of the evolving state vectors, which appears in the standard time-dependent Schrödinger equation, becomes ill-defined. A proper formulation of these quantum systems requires the use of a global covariant Schrödinger equation. The local realization of this equation reproduces the usual Schrödinger equation in which the Hamiltonian operator consists of a geometric part and an observable part. If the latter vanishes, the dynamics of the system is time-reparameterization-invariant.

In the textbook treatment of quantum mechanics, one ignores the geometric part of the local Hamiltonian. This is admissible provided that one can remove the geometric part via a gauge transformation. In general, however, the corresponding connection one-form (operator-valued gauge field) may be non-flat, in which case such a gauge transformation does not exist. This observation provides ample motivation for exploring the physical meaning and importance of the corresponding operator-valued curvature two-form.

The splitting of the local Hamiltonians into their geometric and observable parts provides a natural explanation for the differences between the transformation properties of the Hamiltonian and the observables under time-dependent unitary transformations. The existence of a classical analog of this phenomenon, namely the well-known distinction between the transformation properties of a classical Hamiltonian and classical observables under time-dependent canonical transformations, calls for a study of the classical counterpart of the geometric structures we have employed to deal with quantum systems with a time-dependent state space.

As an application of the general results of the present article to a specific example, one can consider the two-level system associated with a Hermitian vector bundle over a sphere that has two-dimensional fibers and an arbitrary metric compatible connection. In Ref. [40] we give a detailed description of this system and derive explicit formulas for its local Hermitian Hamiltonians. A more important area of the application of our results is in the study of quantum systems defined on cosmological backgrounds.

## Data Availability

No new data were produced or analyzed in support of this research.

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
