# Peer review of "Consistent Treatment of Quantum Systems with a Time-Dependent Hilbert Space"

_entropy, 2024, doi:10.3390/e26040314_

Round 1

Reviewer 1 Report

Comments and Suggestions for Authors

In this manuscript ‘Consistent treatment of quantum systems with a time-dependent Hilbert space’, the author focuses on an important fundamental aspect of quantum mechanics. In details, starting from the definition and misunderstanding of ‘Hermitian’ or ‘Hermiticity’, which leading to the observables to be not invariant after unitary transformations, the author introduces his thinking and solutions, i.e., a new representation related to time-dependent Hilbert spaces and Hamiltonians.

The author defines new concepts and introduces new equations, such as, covariant time derivative, covariant Schrodinger equation, time-dependent inner products, etc., and reveals the geometric part and observable part of Hamiltonian operators.

In all, I believe the mathematical derivations in it are correct in general, and especially, I believe this article contains enough novelties and will attract wider readers who are interested in foundations of quantum physics. Therefore, I approve its directly publication in ‘Entropy’ after considering my following points:  

1)    In line 92 on page 3, whether the \alpha is normalized? If not, does it mean that your theory hereafter is suitable for both Hermitian and NH quantum systems? If so, your theory maybe treated as much wider quantum theory for both the conventional and NH quantum mechanics, and I suggest to add some texts about open quantum systems in the introduction part by adding (not required):

[r1] Breuer,H.P. & Petruccione,F.The Theory of Open Quantum Systems.10th anniversary ed.(OxfordUniversityPress,Oxford,2002).Barreiro,J.T.etal.Anopen-system quantum simulator with trapped ions.Nature470,486–491(2011).

[r2] Hu, Z., Xia, R. & Kais, S. A quantum algorithm for evolving open quantum dynamics on quantum computing devices. Sci. Rep. 10, 3301 (2020).

[r3] Del Re, L., Rost, B., Kemper, A. F. & Freericks, J. K. Driven-dissipative quantum mechanics on a lattice: Simulating a fermionic reservoir on a quantum computer. Phys. Rev. B 102, 125112 (2020).

[r4] Zheng, C. Universal quantum simulation of singlequbit nonunitary operators using duality quantum algorithm. Scientific Reports (2021) 11:3960

2)    I believe that if you can focus more on the physical meanings of your theory, it will be more attractive and easier to be understand for readers with physics backgrounds, e.g., is it possible to analyze the physical meaning of covariant time derivative D_t of Eq. (17)?  

3)    There is a math notion of ^ in equations (41) and (42), and I think it is better to the general readers if the author can give the definition or a brief statement.

4)    When I read the 4th part, the first impression is that it can be treat as a time-dependent generalization of the author’s previous theory of inner product in pseudo-Hermitian systems. However, it is a little conflict to the author’s statement that ‘n this sense referring to (54) as time-dependent pseudo-Hermiticity or quasi-Hermiticity condition is misleading, for it does not impose any condition on the Hamiltonian.’ In the last line of part 4 (line 518 and 519). Could the author give me more explanations?

5) Some typos, such as ‘… observable that …’ in line 83, minisuperspace in line 165.

Author Response

Comments of Referee 1: This referee provides a brief summary of the subject and results of the paper and writes: “I believe this article contains enough novelties and will attract wider readers who are interested in foundations of quantum physics. Therefore, I approve its directly publication in ‘Entropy’ after considering my following points:” In the following I summarize these points and respond to them in the same order as they appear in the referee report.

Comment 1: Referee 1 asks whether the \alpha in line 92 on page 3 is normalized, “if not, does it mean that your theory hereafter is suitable for both Hermitian and NH quantum systems? If so, your theory maybe treated as much wider quantum theory for both the conventional and NH quantum mechanics”. (S)He then suggest that I “add some texts about open quantum systems in the introduction part” and cite some references to open quantum systems.

Author’s Response to Comment 1: I think that there is a misunderstanding regarding the meaning and role of alpha in line 92 on page 3. By definition, the states of a quantum system are rays (one-dimensional subspaces) of the Hilbert space. Therefore, any two state vectors defining the same state can at most differ by a multiplicative constant. This constant can be any nonzero complex number, because the dimension of the state (ray) is one.

Regarding open quantum systems, I would like to stress that in this article I address some basic problems related to closed quantum systems defined on a time-dependent Hilbert space where all the axioms of quantum mechanics are enforced. The treatment of open quantum systems requires effective descriptions that in my opinion should be dealt with separately. In particular, I do not believe reference to open quantum systems can in any way improve the presentation of the results of this paper.

Comment 2: The referee writes: “I believe that if you can focus more on the physical meanings of your theory, it will be more attractive and easier to be understand for readers with physics backgrounds, e.g., is it possible to analyze the physical meaning of covariant time derivative D_t of Eq. (17)?”

Author’s Response to Comment 2: The notion of covaraint time derivative that I employ is identical to the one known in gauge theories. Its physical importance becomes clear when I show how it affects the dynamics of the system in terms of the geometric part of the local Hamiltonian. The main motivation for introducing covaraint time derivative (17) is the need for a meaningful notion of the ``time derivative’’ of evolving state vectors that belong to a Hermitian vector bundle. I think I have provided sufficient motivation for the main problems I address in this paper. The solution for these problems however require the use of certain mathematical tools like covariant derivative, connection, vector bundle, etc. I have cited the relevant books covering this material that are primary intended for physicists. I do not think I can provide an elementary exposition of these tools within the limitations of a research article.

Comment 3: The referee writes: “There is a math notion of ^ in equations (41) and (42), and I think it is better to the general readers if the author can give the definition or a brief statement.”

Response to Comment 4: The wedge symbol ^ stands for the standard wedge product of differential forms. I have added a footnote (Footnote 5) where I state this.

Comment 4: The referee writes: “When I read the 4th part, the first impression is that it can be treat as a time-dependent generalization of the author’s previous theory of inner product in pseudo-Hermitian systems. However, it is a little conflict to the author’s statement that ‘n this sense referring to (54) as time-dependent pseudo-Hermiticity or quasi-Hermiticity condition is misleading, for it does not impose any condition on the Hamiltonian.’ In the last line of part 4 (line 518 and 519). Could the author give me more explanations?”

Author’s Response to Comment 4: I have rephrased the content of the last paragraph of Section 4 to clarify the situation and added a separate remark to reveal the relevance of the results of this section to those of Section 3.

Comment 5: The referee asks that I correct the typos on lines 83 and 165.

Author’s Response to Comment 5: I have corrected the mentioned typos.

Reviewer 2 Report

Comments and Suggestions for Authors

This is a nice, almost pedagogical review of this area.  It does need to be checked for typos, which are numerous, e.g. "Schrodiger" equation on ln. 308, an apparently incorrect index 1 in the unnumbered equation after ln. 414, many others.

Author Response

Comments of Referee 2: This referee writes: “This is a nice, almost pedagogical review of this area.  It does need to be checked for typos, …”

Author’s Response: I have located and corrected the typos in the paper.

Reviewer 3 Report

Comments and Suggestions for Authors

This article is written clearly, it is   mathematically rigorously  and correct. This work is devoted to the study of the problem of extending canonical quantum mechanics to the case of time-dependent Hilbert spaces. The manuscript clearly demonstrates how in this case such systems can be described in terms of time-independent Hilbert space. In this case, however, it is necessary to sacrifice the canonical definition of an observable in quantum mechanics, since a self-adjoint operator may not be an observable, in particular,  the Hamiltonian operator.

This article is very important because it fills a colossal gap in generalizing the methods and results of canonical quantum mechanics. Therefore, I recommend it for publication in  Entropy. However, I have one important note. This work is essentially mathematical, as it is devoted to expanding the mathematical foundations of quantum mechanics, but its methods are intended for real physical systems.  The article lacks a specific example of a real physical system where the results obtained can be applied. In my opinion, the author should give such an example. In this case, of course, there is no need to study such a system in detail. Just give an example and a short explanation.

Author Response

Comments of Referee 3: After praising the article by indicating that “This article is written clearly, it is mathematically rigorously  and correct.”, Referee 3 provides a brief summary of its results and writes: “This article is very important because it fills a colossal gap in generalizing the methods and results of canonical quantum mechanics. Therefore, I recommend it for publication in  Entropy.” He then suggests that I include a brief discussion of the application of the results for “a specific example of a real physical system.” 

Author’s Response: I have spent considerable time to come up with a specific physical system to which I can apply the general results of this article. The simplest example that I could find was a two-level quantum system where the base space M for the corresponding Hermitian vector bundle is a two-dimensional sphere. I do not include the details of the calculations for this example, because they are given in my first paper on this subject, namely Ref. 39. In last paragraph of Section 5 of the revised manuscript I make an explicit comment on this example and suggest that the results of this paper can have useful applications in the study of quantum systems defined on cosmological backgrounds.

I would like to emphasize that the aim of the present article is to draw the reader’s attention to certain basic open problems of quantum mechanics and to outline a geometrical approach to deal with them. I believe that the current version of the manuscript serves this purpose satisfactorily. A comprehensive study of the conceptual implications and physical applications of this approach is a subject for future research. I hope that this work serves as an invitation to engage in this line of research.

Reviewer 4 Report

Comments and Suggestions for Authors

The authors provide an in-depth analysis of quantum systems with a time-dependent Hilbert space and potential. They convincingly show that these systems do, indeed, require much more attention than received in QM textbooks. The authors demonstrate that the time-derivative in Schrödinger’s Equation (SE) can become ill-defined when the state-vectors undergo non-trivial evolutions. Relatedly, the authors formulate the general form of the required quantum system in which the standard SE is generalized to a globally covariant SE. Relatedly, the authors show that the Hamiltonian operator must generalizes to have both a gauge-like geometric part and an observable part. The paper is well-written and well-organized. It definitely warrants publication. However, the main suggestion I have before publication, is that it might be worthwhile for the authors to provide worked out example cases for the reader. In particular, both a more detailed one-parameter case (beyond the present brief discussion of a varying width infinite square-well potential), and a multi-parameter case could be shown. Since standard textbooks do present the geometric aspects of a time-independent potential in the form of Barry’s phase, the authors could then explicitly generalize Barry’s phase for time-evolving potentials. This would be very worthwhile for the reader. Further, such examples would resolve the one issue I noted as “can be improved”, clarifying the methodology for deriving the covariant SE. 

Author Response

Referee 4’s comments are in agreement with those of Referee 3. S(H)e also makes a number of positive remarks on the content of the paper and writes: “The paper is well-written and well-organized. It definitely warrants publication.” His/Her only suggestion is that “it might be worthwhile for the authors to provide worked out example cases.” 

Author’s Response: I have spent considerable time to come up with a specific physical system to which I can apply the general results of this article. The simplest example that I could find was a two-level quantum system where the base space M for the corresponding Hermitian vector bundle is a two-dimensional sphere. I do not include the details of the calculations for this example, because they are given in my first paper on this subject, namely Ref. 39. In last paragraph of Section 5 of the revised manuscript I make an explicit comment on this example and suggest that the results of this paper can have useful applications in the study of quantum systems defined on cosmological backgrounds.

I would like to emphasize that the aim of the present article is to draw the reader’s attention to certain basic open problems of quantum mechanics and to outline a geometrical approach to deal with them. I believe that the current version of the manuscript serves this purpose satisfactorily. A comprehensive study of the conceptual implications and physical applications of this approach is a subject for future research. I hope that this work serves as an invitation to engage in this line of research.